# Endosseous Dental Implant Materials and Clinical Outcomes of Different Alloys: A Systematic Review

**DOI:** 10.3390/ma15051979

**Published:** 2022-03-07

**Authors:** Luca Fiorillo, Marco Cicciù, Tolga Fikret Tozum, Matteo Saccucci, Cristiano Orlando, Giovanni Luca Romano, Cesare D’Amico, Gabriele Cervino

**Affiliations:** 1Department of Biomedical and Dental Sciences and Morphological and Functional Imaging, Messina University, 98100 Messina, Italy; lfiorillo@unime.it (L.F.); cdamico@unime.it (C.D.); gcervino@unime.it (G.C.); 2Multidisciplinary Department of Medical-Surgical and Dental Specialties, Second University of Naples, 80100 Naples, Italy; 3Department of Dentistry, University of Aldent, 1000 Tirana, Albania; 4Department of Periodontics, College of Dentistry, University of Illinois at Chicago, Chicago, IL 6007, USA; ttozum@uic.edu; 5Department of Oral and Maxillofacial Sciences, Sapienza University of Rome, 00161 Rome, Italy; matteo.saccucci@uniroma1.it; 6Behavioural Health Institute, Loma Linda University, Loma Linda, CA 92354, USA; orlandocristiano95@gmail.com; 7Department of Biomedical and Biotechnological Sciences (BIOMETEC), Section of Pharmacology, University of Catania, 95124 Catania, Italy; giovanniluca.romano@unict.it

**Keywords:** dental implants, dental materials, titanium dental implant, zirconia dental implant, dental implant alloys

## Abstract

In recent years, implantology has made significant progress, as it has now become a safe and predictable practice. The development of new geometries, primary and secondary, of new surfaces and alloys, has made this possible. The purpose of this review is to analyze the different alloys present on the market, such as that in zirconia, and evaluate their clinical differences with those most commonly used, such as those in grade IV titanium. The review, conducted on major scientific databases such as Scopus, PubMed, Web of Science and MDPI yielded a startling number of 305 results. After the application of the filters and the evaluation of the results in the review, only 10 Randomized Clinical Trials (RCTs) were included. Multiple outcomes were considered, such as Marginal Bone Level (MBL), Bleeding on Probing (BoP), Survival Rate, Success Rate and parameters related to aesthetic and prosthetic factors. There are currently no statistically significant differences between the use of zirconia implants and titanium implants, neither for fixed prosthetic restorations nor for overdenture restorations. Only the cases reported complain about the rigidity and, therefore, the possibility of fracture of the zirconium. Certainly the continuous improvement in these materials will ensure that they could be used safely while maintaining their high aesthetic performance.

## 1. Introduction

### 1.1. Background

Dental implant is a sector of dentistry that has developed considerable progress in recent years. The main bases of osseointegrated implantology have undergone significant modifications that are altering the treatment approach to the daily practice [1].

In fact, the field has been able to achieve excellent results in oral rehabilitation, solving aesthetic, functional and psychological problems in both partial and total edentulism [2,3]. In recent years, the literature has focused on understanding what the gold standard is for implant surfaces in terms of both speed of osseointegration and the amount of bone–implant bonding [4,5].

There are currently a number of titanium dental implants on the market whose surface is subjected to different treatments. These are able to conditionate the surface characteristics in order to improve osteoconductivity. Starting by the smooth or machined surface implant used as the first prototype, clinicians and researchers have moved on to dental implants structured with macro- and micro-topography by means of sandblasting procedures with jets of microparticles of aluminum oxide or titanium; high-pressure, high-temperature titanium spray procedures (TPS); high-temperature calcium and phosphate granule melting processes; associated sandblasting and etching with different acids processes; and coating processes with calcium and sodium phosphosilicate glass (bioactive glass) [6].

Osseointegration is defined as:

“*direct contact between the vital bone tissue and the surface of a dental implant, without the interposition of soft tissue [1]*”.

The basic requirements for successful osseointegration were codified by Branemark, Alberktsson et al. in the early 1980s [7]. The positioning of dental implants within the bone structure is considered as an impairment of the totality of the organism. Around an implant, there is always a thin empty layer in which complex biological phenomena take place. All the phases of inflammation take place, which inevitably leads to the reabsorption of the cortical bone of the implant preparation canal and, subsequently, to the new ossification with osseointegration of the implant fixture. Precisely, immediately, there is tissue ischemia with necrosis of the surrounded bone tissue side [8]. The increase in vascular permeability in the zone of the operation determines the pouring of undifferentiated mesenchymal cells to fill the gap between the bone surface and the implant (cellular and vascular processes). After the first days, cell differentiation and organization of the peri-implant tissue occur to allow removal of necrotic or damaged tissues by macrophages to begin the reparative phase.

The bone neoformation process adheres to all the stages that influence direct ossification: arrival of bone cells, deposition of osteoid tissue and formation of immature bone with interwoven fibers. The immature bone is then resorbed and substituted with mature lamellar bone; this event then guides the formation of new bone around the fixture [9,10,11] (Figure 1).

Several components have been tried out in the manufacturing of dental implants, some of which (Cr, Ni, V; steel; Cr, Co, Mo; Au, Pl, Ag, Cu) are no longer used. Implant materials considered biocompatible today are Ti cp (commercially pure titanium) and titanium alloys such as (Ti_6_Al_4_V), Al_2_O_3_-based ceramics, hydroxyapatite and zirconia [12,13,14,15,16].

#### 1.1.1. Titanium

Titanium is a particularly popular metal in dentistry; the mechanical–physical and biological properties of titanium allow it to be used commonly for prosthodontics’ components, skeletons and implant structural devices [17,18].

Titanium has a relative density of 4.5 g/cm^3^, melts at 1677 °C and boils at 3277 °C; it has a low thermal conductivity (22 Wm^−1^ K^−1^). It has a high mechanical strength (tensile strength of 240 MPa) with an elongation at break of 12%. The modulus of elasticity is relatively low and similar to that of bone (110,000 N/cm^2^) [19]. The fundamental characteristics of this metal are its high protection to corrosion and its high degree of biocompatibility. In nature, it is found generally as a dioxide (TiO_2_) in three polymorphic conversions: rutile, anatase and brookite [20,21].

#### 1.1.2. Zirconia

Zirconia, instead, looks like an odorless white powder at room temperature. It is a polymorphic crystalline substance. Up to 1170 °C, it has a monoclinic crystal structure. From 1170 °C to 2370 °C, it has a tetragonal crystal structure. Above 2370 °C, it has a cubic crystal structure. The melting point is 2680 °C. During heating, the transformation from the monoclinic structure to the tetragonal structure occurs with a volumetric contraction of about 5% (tensile strength of 3895.2 ± 202.9 MPa) [22]. Zirconium oxide has a high biocompatibility and antimicrobial activity; it is in fact used in different biomedical fields for specific applications that fall within the dental and orthopedic field, in which the material is in direct contact with biological tissues. In particular, zirconium oxide is defined as a bio-inert ceramic material: the material does not induce or undergo chemical or biological alterations due to contact with the biological environment. Given the excellent properties of nanostructured zirconium, ZrO_2_ nanoparticles are used both as a ceramic coating of titanium metal implants or for the construction of entire dental implants. In the studies carried out, particular attention was paid to the effect of modification of the zirconium oxide surfaces on osseointegration, and it was shown that subtle surface changes in the zirconium oxide had a strong impact on the bone application of the implant surface [23].

### 1.2. Aim

The null hypothesis of this review is that zirconia dental implants offer better clinical parameters in rehabilitation. Therefore, the main aim of this study is to analyze whether different dental implant materials or alloys could modify clinical outcomes in dental implant rehabilitation (Section 2.4), referring to titanium dental implants alloys and zirconia dental implants. The PICO question is:Does zirconia alloy dental implant have different success rates in patients who undergo dental implant rehabilitation over time vs. titanium alloy dental implants?

## 2. Materials and Methods

### 2.1. Eligibility Criteria

This review has been conducted according to the PRISMA checklist. First of all, the authors stated “eligibility criteria”, parameters that had to be respected in order to be included in the results of the review.

The inclusion criteria are as follows:Studies about patients who need implant–prosthetic rehabilitation therapy,Studies that compare titanium alloys/zirconia dental implants; randomized clinical trials (RCT),Studies on humans,Studies concerning completed implant–prosthetic rehabilitation,

Exclusion criteria:Studies about dental implant rehabilitation in patients with systemic disease,Studies about dental implant rehabilitation in patients with local or systemic contraindication to dental implant therapy,Not on human studies,Unpublished or ongoing studies,Studies older than 15 years.

### 2.2. Information Sources

The research sources are the most common and reliable scientific research channels according to the scientific community:Scopus ElsevierWeb of ScienceGoogle ScholarPubMedMDPI database

### 2.3. Search Strategy

The search strategy was to insert the following word string in the information sources, according to the previous paragraphs:

“Zirconia implants”; “Zr implants”; “zirconia dental implants”; “Zr implants”; “zirconium implants”; “Zirconium dental implants”.

In the search strategy, it was not considered to include the word “titanium” (concerning titanium implants), in order to obtain only RCTs that would compare zirconia vs. titanium implants. Furthermore, recent works were considered, in accordance with inclusion and exclusion criteria, and only some types of studies were selected, using the filters already present in the scientific information sources.

Subsequently, once the studies were obtained, they were subjected to a first screening through independent reading by the authors of “title” and “abstract”, and then the full text was read (logically, if they were not discarded earlier, as they did not meet the inclusion criteria already when reading “title” or “abstract”).

The revision method is in accordance with the PRISMA statement, with methodology and subdivision in the manuscript according to their parameters, as well as with the presence of the PRISMA flow chart [24,25,26].

As regards the identification of the study objective, it also made use of the PICO—Population Intervention Comparison Outcome system [27], as specified in the Introduction section.

### 2.4. Data Items

Main data outcome and evaluated items are showed here:Clinical:○Implant related:▪Implant survival▪Implant success○Periodontal:▪Plaque index (PI)▪Full mouth plaque score (FMPS),▪Modified sulcus bleeding index (SBI)▪Bleeding on probing (BOP)○Prosthetic:
▪Prosthetic events▪Prosthetic success○Aesthetical:
▪Pink esthetic score (PES)▪Aesthetic outcome○Unobjective data:
▪Condition of the peri-implant mucosa▪Radiographical:▪Marginal bone loss (MBL)▪Patient related:▪Visual Analogue Scale (VAS)

### 2.5. Study Risk of Bias Assessment

The risk of bias was assessed according to the studies and method proposed by Cochrane and Higgins et al [28,29,30,31,32]. The risk of bias between the studies was assessed; in addition, as there were randomized clinical trials, the level of bias within the individual studies was assessed. Randomized trials and systematic reviews of such studies provide highly reliable evidence as to the causes of health interventions. Provided there are substantial participants, randomization should guarantee that contributors in the intervention and comparison groups are similar with respect to known and unknown prognostic factors. The tool provides for the assignment of a high, low or unclear risk judgment of material bias for each element. We define material bias as bias of sufficient magnitude to have a noticeable effect on the study results or conclusions, recognizing the subjectivity of that judgment. The specific criteria for making judgments on the risk of bias from each of the elements of the tool are available in the Cochrane Handbook [32]. Some studies taken into consideration may present an incompleteness of the reported data; this may introduce a certain bias to the results of the review. For example, in the event that some subjects of the RCTs considered do not complete the study (in this case, rehabilitation or follow-up), attrition bias is introduced.

To summarize conclusions about the overall risk of bias within or between studies, it is fundamental to underline the evaluations between items in the tool for each outcome within each study.

### 2.6. Synthesis Methods

The studies were read by the authors individually and were subsequently discussed and analyzed. Each author created tables and cards with the main results of all the articles included; these were reviewed and are exposed in the “results” and “discussion” sections.

## 3. Results

### 3.1. Study Selection

The first search on information sources according to the Materials and Methods section reported a total of 305 results and, after applying filters, a total of 18 results was obtained. After a first read of the abstract and later of the full text to evaluate inclusion and exclusion criteria, only 10 studies were selected to be included in this review (Figure 2).

### 3.2. Study Characteristics

Studies’ characteristics are showed in Table 1, as follows:Authors—First author study nameYear—Year of publicationMethodology—Presence of blinding/allocation/and other methodologySample—Sample type and numberFollow-up—Follow-up time

### 3.3. Results of Individual Studies

Results of individual studies have been collected and are shown in Table 2, according to these parameters:Authors—First author study nameGroups—RCT groups subdivision (subdivided by semicolon)Outcomes—Main outcomes evaluation (subdivided by lines)Main results—Results related to “outcomes” column (subdivided by lines respectively to Groups)

### 3.4. Results of Syntheses

Bienz et al. [9], in their study, evaluated differences between zirconia and titanium dental implants. In 42 patients with two adjacent missing teeth, they randomly positioned one zirconia and one titanium dental implant. After 12 weeks, half of the sample was instructed to omit oral hygiene for three weeks, and then the authors evaluated some clinical parameters, as shown in paragraph 3.4. The plaque control index increased significantly in both dental implants groups, with significant difference in favor of zirconia dental implants. The BoP index was higher in titanium dental implants, as well; however, there were no histologically significant differences between groups regarding inflammatory cells number and junctional epithelium. Patil et al. [33] performed a RCT study to evaluate crestal bone level changes in patients with mandibular overdenture retained by one or two zirconium–titanium dental implants with one year of follow-up. In a sample of 24 participants, 36 dental implants were placed by a single operator. Dental implants were immediately connected through LOCATOR attachments to the overdenture. The authors evaluated bone loss at one month and one year, with no significant differences between groups. Furthermore, they evaluated patients’ satisfaction in both groups before rehabilitation, at one month and one year’s time; these data increased over time but did not show significant differences between groups. Koller et al. [34], in their study, evaluated clinical and radiographical differences between titanium and zirconia dental implants. These biphasic dental implants were evaluated on plaque index, bleeding on probing, pink esthetic score and marginal bone loss. In both groups, 31 dental implants were inserted, but 3 of these failed (2 zirconia and 1 titanium), so 28 implants (14 zirconia and 14 titanium) were evaluated on 21 healthy patients with a mean of 80.9 months of follow-up. No significant statistical differences were shown by the authors, except the intergroup BoP at 30 months in favor of zirconia dental implants *p* = 0.001. Payer et al. [35] evaluated the outcomes (both clinical and radiographic) of two-piece zirconia dental implants vs. titanium implants over a period of 24 months of follow-up. In this study, 31 dental implants of the same manufacturer (16 zirconia and 15 titanium) were placed in 22 patients. After six months of healing for the upper jaw and four months for the lower jaw, dental implants were loaded. The authors evaluated radiographic bone loss, implant survival and success, and condition of mucosa. All evaluated data did not show statistical differences between groups. Müller et al. [36], in their five-year follow-up study, evaluated the differences between titanium–zirconium dental implants and titanium grade IV dental implants in retaining mandibular overdentures. They recruited 91 patients, but only 47 of these completed the follow-up time. Dental implants were inserted randomly in the interforaminal region and crestal bone level and success rate and survival rate were evaluated. There were no significant differences in crestal bone level and success or survival rate between groups. Ioannidis et al. [37], in their study, evaluated the differences between the use of titanium–zirconium narrow dental implants versus the use of titanium implants in respect to marginal bone and clinical parameters. In this study, 40 patients where initially included; these patients needed a single tooth restoration that was performed with a randomly assigned dental implant and a porcelain-fused-to-metal crown six months after surgery. After 3 years of follow-up, only 32 of the 40 patients were examined, with no implant failures and no significant differences with respect to MBL, papilla levels and success rate. The only recorded statistical difference was with respect to FMPS in favor of zirconium dental implants (but a full mouth parameter should not be related to a single dental implant restoration). The study of Osman et al. [38] showed how there could be a difference between these two types of alloys for dental implants. They evaluated 129 dental implants in a 24 patient sample. Dental implants were randomly allocated by a monophasic titanium dental implant vs. a monophasic zirconia dental implant, both used to perform mandibular and maxillary overdentures. The authors respected a conventional loading protocol and evaluated survival rate and MBL over time. There was no significant difference in survival rate between groups. It is useful to report that three implants in the zirconia group fractured. A significant value of marginal bone loss was recorded, and titanium dental implants presented a lower value; furthermore, the prediction model revealed a higher risk of implant failure in the maxilla (*p* < 0.0001). Osman et al. [39], in their 2014 preliminary study, evaluated yttrium-stabilized tetragonal polycrystalline zirconia vs. titanium grade 4 dental implants, not by a surgery and periodontal point of view, but by a prosthodontic point of view. After 3 deaths and 2 drop-outs, only 19 participants were evaluated; there were 79 maintenance prosthodontic events and 3 zirconia dental implant fractures (one mandibular and 3 maxillaries were recorded). There were no significant differences in maintenance between groups. Al-Nawas [40] performed a double-blind, split-mouth RCT with a sample of 91 patients with a year follow-up. They experienced no differences between success rate and marginal bone level changes during the follow-up period, and furthermore, they evidenced no differences in survival rate, PI and SBI. In this split-mouth study, according to the authors, zirconium–titanium alloy dental implants were performed as titanium grade IV dental implants. The authors evaluated population safety, reporting only 37 adverse events in the course of the study and 7 severe adverse events, but not related to the study device. Cannizzaro et al. [41] performed a study evaluating differences in protocol loading referred to zirconia dental implants. They evaluated differences in immediately placed non-occlusally loaded crowns vs. immediately occlusally loaded crowns on zirconia dental implants. Dental implant success rate was evaluated, along with any biological or prosthetic complications and marginal bone levels over time. Results did not show significant differences in these outcomes; the only fact to report was about immediate post-extractive implants: 40% of failure in this subgroup vs. 3% in the delayed group (*p* = 0.010).

### 3.5. Reporting Biases

Risk of bias has been evaluated according to the Materials and Methods section (Table 1), and it is shown in Table 3.

In Müller et al.’s [36] study, due to the indistinguishable nature of the dental implants used by the operators, it was possible to further reduce the risk of bias. In this case, in fact, the titanium implants were indistinguishable from those in the titanium–zirconium alloy. For this reason, the dental implants were inserted in a split-mouth manner by blinded operators. The same could be said for Al-Nawas et al.’s [40] study. Unfortunately, some of the included studies present selective reporting. This is a major limitation, as the results of some outcomes are not present in the form of raw data, or at most, the statistics are present or the latter is missing. In particular, selective reporting has been highlighted in the studies of Payer et al. [35], Osman et al. [39] and Cannizzaro et al. [41]. Fortunately, however, one thing that everyone agrees on is the random generation of groups, thus guaranteeing one random sequence generation, the “must have” of a randomized clinical trial.

### 3.6. Additional Analysis

Further analysis on obtained results was performed to better evaluate rough data (Figure 2). In this figure, difference over time in MBL is shown; furthermore, there are no statistical differences between zirconia dental implants and titanium dental implants on survival rate. This is highlighted as a linear trend line which is higher for zirconia alloy dental implants. For this figure’s realization, only the studies showing MBL as an outcome (Patil et al. [33], Koller et al. [34], Müller et al. [36], Ioannidis et al. [37], Osman et al. [38,39], Al-Nawas [40] and Cannizzaro et al. [41]) were taken into consideration; the data were processed starting from the raw data, and we obtained weighted mathematical averages for the realization of Figure 2. As explained later, it is not possible to realize a meta-analysis, as many data are missing in some studies, are evaluated differently or many heterogeneous results were present or not available as raw data. Forcing a statistical calculation would introduce a risk of bias in the results (Figure 3).

## 4. Discussion

Implant surface treatments aim to produce a biologically active surface that improves osseointegration between implant and bone tissue. With threading alone, the degree of resistance to tensile and compressive forces is greater than with smooth implants; the presence of micro retention on the surface of the fixture makes it possible to increase the tensile and torsional strength of the implant. Some contributors have highlighted that macrophages, epithelial cells and osteoblasts have a large trophism towards rough surfaces: “rugophilia” [42]. After analyzing the results regarding the systematic review, the evolution of the materials and titanium alloys used for the realization of the dental implants over time will be briefly described.

In order to obtain an ideal roughness that facilitates osseointegration, different techniques of micro treatment of the external surface have been proposed over the years [43].

### 4.1. Dental Implant Alloys and Surface Treatments

Other techniques include coating with bioactive material, such as biocompatible glass.

The different types of dental fixture surfaces can be divided into two main sections:Smooth implants andRough implants:
○rough implants by addition of material (TPS, coating with ceramic materials such as hydroxyapatite or bioactive glass) and○rough implants by subtraction of material (sandblasted and/or etched) [44].

#### 4.1.1. Smooth Surfaces

Smooth implants are electropolished or machined. For electropolished implants, the surface is treated by conditioning in an electrolyte solution in which current passes and where the other electrode is represented by platinum. This treatment could achieve a roughness value of 10 nm. For machined or turned implants, the surface is mechanically turned; microscopically, it appears shiny and polished, while a scanning microscope could highlight turning striations with minimal roughness.

Clinical and in vitro research has underlined the preponderance of rough-surface implants over smooth-surface implants in terms of the quick bone integration phase, the percentage of healing at the bone–implant interface and the resistance to torsion proofs [45].

The development of the titanium has made it possible to partially halve the healing and prosthetic protocol times of implants from six months to three months or less. Due to the considerable hydrophilicity and, consequently, the greater ability to attract organic liquids such as blood. In this connection, mention should be made of the work of Johnson Davies, wherein platelet activation increases in rough surfaces, with a consequently greater presence of molecular mediators, and fibrin retention on the surface of the implant increases, thus leading to a stronger fibrin network, which is already present in the initial phases of the osseointegration and is basilar for the bone–implant healing. In smooth implants, on the other hand, platelets are lower, the fibrin is less strongly attached and the pericyte cells are not able to reach the surface of the implant, giving rise to osteogenesis at a distance and not osteogenesis by contact, as occurs with rough-surface implants [46,47].

#### 4.1.2. Sandblasting

Sandblasting is a method of preparing titanium that gives it a certain surface roughness, obtaining a general improvement in the biomechanical characteristics of the implant.

It is performed by special machines—sandblasting machines—using a high-pressure flow of certain oxides such as aluminum or titanium. The jet, directed onto a clean surface, slightly erodes—its furrows are created with a diameter of about 5–20 m—making it rough. This treatment has a positive influence on primary stability, as macrophages, epithelial cells and osteoblasts show rugophilia for this layer [48].

Sandblasting with aluminum oxide particles with a diameter of 100 to 150 microns has been shown to be compatible with successful bone integration; the irregular surface increases the osteoconductivity of the metal—which is otherwise inert—promoting adhesion and activity of the osteoblasts. As a consequence, healing of the bone–implant interface is quicker.

#### 4.1.3. TPS Surfaces

Another method involves coating with TPS, i.e., titanium powder—usually in the form of a Ti–6Al–4V alloy.

The plasma-spraying procedure is developed by using an electric arc plasma burner; the plasma is produced between a cooled copper anode and tungsten cathode. In this system, titanium powders with a particle size of 50 to 100 microns adhere to the body of the implant on which they are deposited. The coating thus obtained has a thickness of 0.5–2 mm and an average diameter roughness of around 200 mm [49]. Treatment with TPS thus manages to increase the surface area available in contact with the bone, which is greater than with normal sandblasting, with superior anchorage to the bone tissue. The downside of this procedure is the low control of contamination, the opportunity of particle detachment and the diffusion of metal ions from the implant surface [50].

#### 4.1.4. Hydroxyapatite (HA) Coated Surfaces

Hydroxyapatite is a substance that forms part of the composition of all mineralized structures in the body; as well as being autologous, it can be reproduced industrially with a porous or dense consistency, granular powder and preformed blocks [51]. It binds chemically to the bone (titanium uses a mechanic bond, benefiting from surface micro retention) and does not induce toxicity or inflammation, either local or systemic [52,53].

The insufficient mechanical properties are improved by its application as a coating for titanium surfaces. It is represented by calcium and phosphate granules linked through high-temperature processes (1250 °C). The high concentration of phosphate and calcium should promote migration and maturation of bone tissue cells that conditionate the bone matrix, and it should permit a faster mineralization [54,55]. The processing technique is similar to that of obtaining TPS, with a reduction in the crystallinity of this material (from 5% to 60–70%) [56].

Several studies have shown that HA coating could lead to an improvement in clinical results, with higher bone-to-screw contact, more bone tissue and no areas of osteolysis between the threads of HA-coated screws, compared to uncoated screws [57]. A major clinical problem highlighted by the literature concerns precisely the HA coating of dental implants; in fact, it has been shown that some HA coatings could, after some time, completely detach from the metal surface of the dental implant. Different formulations have been proposed for the HA coating, and it has been shown that the release may depend on the density and on some mechanical characteristics of the HA [58].

#### 4.1.5. Sandblasted and Etched Surfaces (SLA^®^)

In 1990, a new surface called SLA (Sand-blasted Large-grid Acid-etched) was proposed: it is a surface sandblasted with coarse-grained sand (250–500 nm diameter), washed in an ultrasonic tank with deionised water, then dried, placed in a thermostatically controlled solution of hydrochloric and sulphuric acid and finally reworked and re-dried by hot air.

This type of surface treatment produces a macro-rough layer due to the sandblasting process, while at the same time, the action produced by the acid generates the formation of microalveoli (micro-roughness).

Implants with SLA endosseous surfaces achieve greater bone contact than the rougher TPS surfaces.

Stain and Mc Collin confirmed that when placing TPS implants in the jaws of patients, it takes three months to achieve stable osseointegration in Misch class 1-2-3 bone, while in class 4 bone it takes four to six months [59]. If, however, dental fixtures with biomimetic characteristics—such as SLA implants—are inserted, the time is considerably reduced to about six weeks in classes 1-2-3 and twelve weeks for class 4, because of the opportunity of promoting the proliferation and differentiation of cells with osteogenic potential in a shorter time and with better performance changes [57,60].

#### 4.1.6. Bioactive Glass Coated Surfaces

Biologically, active glass belongs to the group of glasses which demonstrates biocompatibility in vivo and in vitro, absence of inflammatory and toxic processes in the presence of ontogenetic precursors and osteoconductive capacity to promote a particular biological bond in the glass–bone tissue contact regions. They are mainly composed of calcium and sodium glass phosphosilicates; 45% of the mass is formed by Si_2_O (vitrifying), 24.5% by N_2_O (melting), 24.4 is CaO (stabilizing) and the remaining percentage by P_2_O_2_ (binding) [57,61].

Bioactive glass is unstable in aqueous solution; its “bioactivity” is attributed to a phenomenon of surface hydrolytic degradation with the release of ions, such as sodium, calcium, silicon, phosphorus, potassium, phosphate, capable of creating a layer with high osteoconductive activity guaranteed by hydroxyapatite crystals, which influences the migration and proliferation of some cells, including osteoblasts. The consequences of this phenomenon are the resorption of the vitreous state and its replacement with newly formed bone within a few months [62]. According to Anderson’s classification, class A includes inert glasses, which leads to the formation of the fibrous tissue interface; classes B and C include very and fairly soluble glasses which are not capable of forming a stable bond with the bone tissue; classes D and E include glasses with controlled surface reactivity, capable of forming a more or less consolidated bond with the bone tissue [63].

Recent lab experimentation has evidenced significant denatures of a coating developed as a sprayed coating of bioactive glass on a titanium alloy (using a plasma-spray process); the result is the total degradation of the layer of glassy coating material and its replacement by bone tissue in direct contact with the titanium implant [64,65].

#### 4.1.7. Trade-Mark Surfaces: TiOblast^®^, Osseotite^®^, TiUnite^®^

The TiOblast^®^ (AstraTech^®^, Dentsply Sirona, Rome, Italy) uses an ablative and compactive process. The improved titanium features, through titanium dioxide granules sandblasting under fixed conditions, generate a plastic alteration of the surface and depressions of regular size and shape (1–5 microns). This provides three times the interconnecting force between bone and implant than could be achieved with a standard implant, with an increase in surface area of approximately 15% [66].

The Osseotite^®^ (Biomax^®^ s.p.a., Rome, Italy) surface is structured by being divided into two sections, one smooth and one etched (HCl/H_2_SO_4_). Studies underline that biacidal etching generates a more uniform surface and, thus, significantly greater bone-to-implant contact than machined surfaces.

Anodic oxidation, which is characteristic of TiUnite^®^ (Nobel Biocare^®^, Switzerland), results in a gradual and controlled increase in the surface TiO_2_ layer and surface roughness in the apical direction. The surface has a porous surface which, together with the increasing roughness, results in an increase in contact area. This results in significantly greater bone-to-implant contact than with turned or machined surfaces, higher removal torque and, thus, greater stability [67].

#### 4.1.8. Novel Surface Treatments: Hyaluronic Acid

The modification of the surface of the dental implants is performed to influence the responses from the tissues and to be biologically active; the goal of surface modification is to immobilize proteins, enzymes or peptides on the surfaces of the device in order to induce tissue-specific responses [68,69,70,71].

In 1904, Pfaundler theorized that calcium binding was a fundamental parameter for bone ossification. It was later discovered that glycosaminoglycans (GAGs) play an important role and that hyaluronic acid increases the proliferation and growth of hydroxyapatite crystals. Hyaluronic acid covalently bound to the titanium surfaces of the implant significantly increases bone growth and results in increased maturity for the interfacial bone [72].

### 4.2. Zirconia Dental Implants

Zirconium has biological and mechanical properties that make it an effective alternative on a clinical and competitive level in such a complex market. The chemical–physical properties should be considered first. Zirconia can present at room pressure in three different crystallographic structures (monocyclic, tetragonal or cubic) depending on the temperature at which it is produced. To avoid transitions from the tetragonal structure to the monoclinic structure when the compound cools, the molecule is stabilized with “doping” oxides, such as Y_2_O_3_, CeO_2_, MgO or CaO. This procedure confers considerable stability against microtraumatism, as well as the subsequent surface treatments to which the implants are subjected [73,74]. These treatments are:Sandblasting: this practice, also called abrasion with airborne particles, has the intent of imparting a roughness to the surface at a microscopic level, as opposed to what is obtained after only making the thread (the implant, in this case, is called “machined”). The process produces a uniform micro-abrasion even on hard ceramic and glass materials, and therefore lends itself well to zirconia. Alumina particles are often used, which, however, can show micro-cracks during impact (as mentioned, uniform abrasion must be ensured) and which, in any case, would contaminate the surface. However, in vitro and animal model studies report positive results in terms of osteoconductivity and containment of inflammation.Etching: the use of an acid (hydrofluoric, nitric, sulphidic) allows the homogeneous treatment of the implant surface and, in addition, overcomes the risk of delamination from stress of the material (even if the risk of releasing unwanted substances remains). The method usually involves a thermal type finishing, although recent techniques that combine sandblasting and etching have been proposed in the search for the substrate that is most favorable to osteoblasts.Polishing: on the contrary, the desired effect with this type of treatment is a perfectly smooth surface, even at a microscopic level. The goal is to benefit the epithelial cells, not in contrast with the blastic elements, but with respect to rough surfaces. Polishing machines use silicon carbide paper and diamond or silica suspensions; the method ensures that the surface chemistry of the material is not altered [75].

According to Bienz et al. [9], both zirconia and titanium dental implants show similar clinical outcomes under healthy conditions and correct oral hygiene. Furthermore, under mucositis conditions or low oral hygiene, a lower plaque record and lower bleeding are detected for zirconia implants. Patil et al. [33] show how titanium–zirconium alloy dental implants can represent a valid alternative for overdenture retaining. Furthermore, the authors demonstrate that there are no significant differences in single-implant-retained overdentures vs. two-implant-retained overdentures in the mandible. Koller et al. [34] introduce that no significant differences are highlightable between zirconia and titanium dental implants at 80 months of follow-up. Payer et al. [35] show that there are no significant differences between zirconia dental implants and titanium ones in merit of clinical and radiographical parameters. Furthermore, they state that the bonded zirconia implant abutment connection appears to be capable over the follow-up of 24 months. In a double blinded, split mouth, multicenter clinical trial of Müller et al. [36], authors conclude that there are no significant differences between titanium grade IV and zirconium–titanium alloy dental implants for mandibular overdenture retaining with a follow-up of 60 months. Ioannidis et al. [37] conclude that zirconium–titanium dental implants used for support of a single crown in the aesthetic zone (anterior to premolar) do not differ from titanium grade IV dental implants regarding their clinical performance over a three-year period. Osman et al. [38] believe that caution before recommendation should be made for the use of monophasic zirconia implants for overdentures. According to the authors, their use should be limited only in patients with an allergy to titanium alloys. This decision has been made primarily due to the higher fracture rate and marginal bone loss recorded. Osman et al. [39] continue in a preliminary study, affirming that removable overdentures can be used in “metal-free” rehabilitation, because zirconia dental implants do not interfere with prosthodontic maintenance or success. This perspective randomized controlled double blind clinical trial of edentulous mandibles performed by Al-Nawas et al. [40] confirms that zirconium–titanium dental implants, considering small diameters, as well, provide at least similar outcomes after one year of follow-up as titanium grade IV dental implants. The authors suggest that improved mechanical properties of zirconium–titanium dental implants could extend implant therapy applications. Cannizzaro et al. [41] demonstrate that immediate non-occlusal loading vs. occlusal loading does not provide clinical differences in zirconia dental implants. The only important fact to report is about post-extraction with immediate loading failure that could reach 40% [23]. It is obvious that a zirconium implant, since it is of a white color, can give better aesthetic performance compared to an implant in titanium or titanium–zirconium alloy. From a prosthetic point of view, the advantage is considerable, as it is not necessary to exploit a lot of thickness to layer the ceramics, colors or opaques to hide the metal. However, it is clear from the results that the mechanical characteristics are superimposable, but the zirconium implant shows a greater exposure to fracture due to the mechanical load. The PES and aesthetic outcomes are better for zirconium implants, while the curves show a greater tendency over time to MBL. We repeat that the outcomes do not always show superimposable characteristics and, above all, the differences are not statistically significant. As far as implant survival and implant success are concerned, the results are comparable.

## 5. Conclusions

The study illustrates how the use of endosseous dental implants in different alloys than those most commonly present on the market appears to be safe. In particular, in this review, the authors highlight clinical differences between implants made of grade IV titanium and zirconium implants. Despite zirconia alloys, dental implants show a higher linear trend average value about marginal bone resorption over time. At present, no significant differences appear to be evident in the clinical, radiographic and prosthetic aspects. Surely, zirconium implants could guarantee better aesthetic performance over time, but at the same time, it is necessary to improve these materials, as they have shown a greater tendency to present fractures.

## Figures and Tables

**Figure 1 materials-15-01979-f001:**
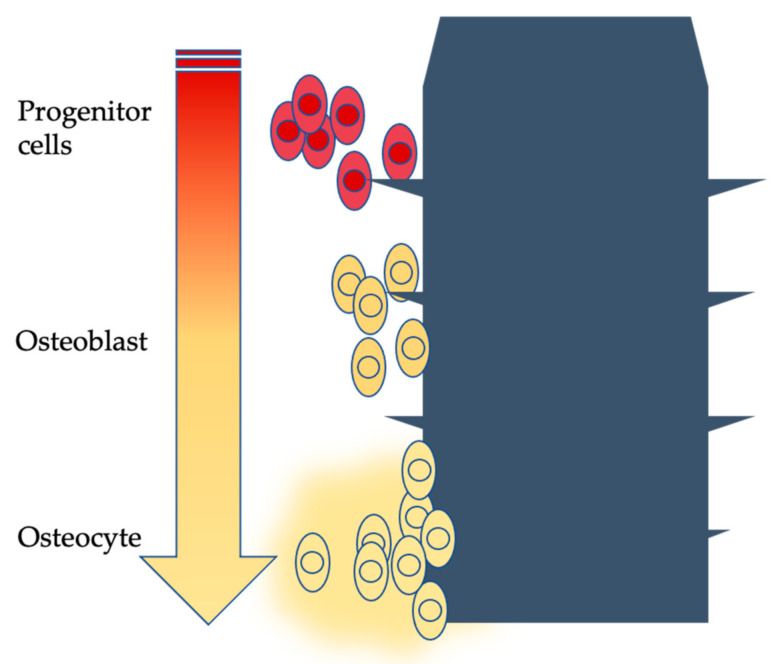
Schematized osseointegration phases over time (arrow).

**Figure 2 materials-15-01979-f002:**
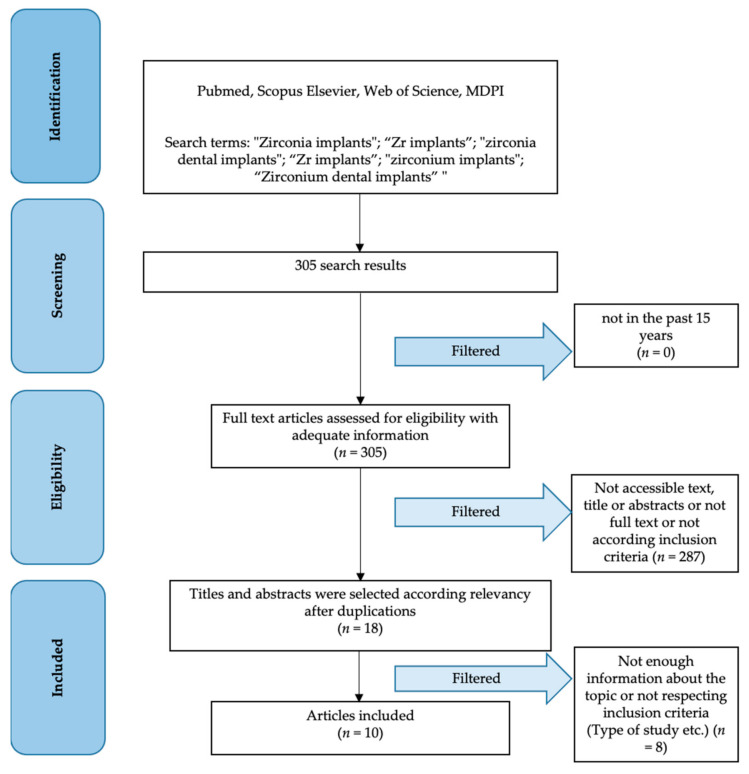
PRISMA flow chart.

**Figure 3 materials-15-01979-f003:**
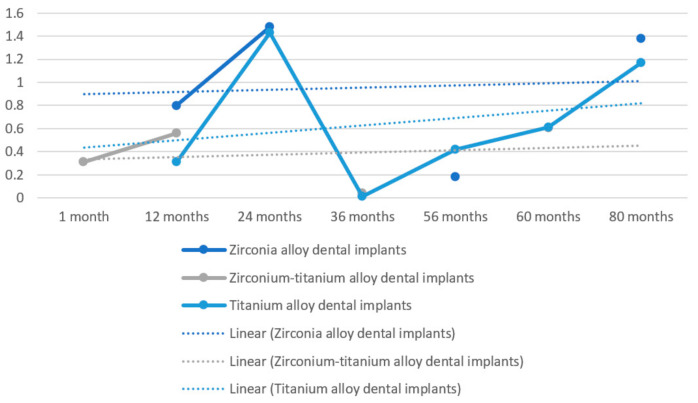
MBL average values over time in zirconia alloy dental implants vs. titanium alloy dental implants vs. zirconium–titanium alloy dental implants according to evaluated study average results. *X* axis: time in months; *Y* axis: millimeters of marginal bone resorption. “Linear” (Lineare) trend line defined for each value (dashed).

**Table 1 materials-15-01979-t001:** Included studies characteristics.

Authors	Year	Methodology	Sample	Follow-Up
Bienz et al. [9]	2021	Randomized dental implant position	42 patients with 84 dental implants	15 weeks
Patil et al. [33]	2020	Blinded statisticians	24 patients with 33 dental implants	1 year
Koller et al. [34]	2020	Random allocation	22 patients with 31 dental implants	80 months
Payer et al. [35]	2015	Random allocation	22 patients with 31 dental implants	24 months
Müller et al. [36]	2015	Double blind/Split mouth	91 patients	5 years
Ioannidis et al. [37]	2015	Random allocation	40 patients with 40 dental implants	3 years
Osman et al. [38]	2014	Random allocation	24 patients with 129 dental implants	56 months
Osman et al. [39]	2014	Random allocation	24 patients with 168 dental implants	1 year
Al-Nawas [40]	2012	Double blind/Split mouth	91 patients with 182 dental implants	1 year
Cannizzaro et al. [41]	2010	Random allocation	40 patients with 40 dental implants	1 year

**Table 2 materials-15-01979-t002:** Main results of individual included studies.

Authors	Groups	Outcomes	Main Results
Bienz et al. [9]	Zirconia dental implant groups vs. Titanium dental implant group; A half with oral hygiene and another one with no oral hygiene for 3 weeks	Plaque control	68.3 ± 31.9% vs. 75.0 ± 29.4%
		(BoP)	21.7 ± 23.6% vs. 32.5 ± 27.8%
		Histology	Number of inflammatory cells not significantly differ
Patil et al. [33]	Single retained overdenture with titanium zirconium dental implant vs. overdenture with 2 titanium zirconium dental implant retention	Crestal bone loss 1 month	0.39 mm vs. 0.23 mm
		Crestal bone loss 1 year	0.88 mm vs. 0.67 mm
		VAS on patients satisfaction 1 month	49.7% vs. 54.8%
		VAS on patients satisfaction 1 year	54.5% vs. 58.9%
Koller et al. [34]	Zirconia dental implants vs. titanium dental implants	PI	11.07% vs. 15.20%
		BoP	16.43% vs. 12.60%
		PES	11.11 vs. 11.56
		MBL	1.38 mm vs. 1.17 mm
Payer et al. [35]	Zirconia dental implants vs. titanium dental implants	Radiographic bone levels	1.48mm vs. 1.43 mm
		BoP	9.1% vs. 7.4%
		PI	19.38 vs. 16.05
		PES	11.22 vs. missing
		Implants stability	–2.5 for all
		Clinical evaluation	Missing vs. 10.75
Müller et al. [36]	Titanium-zirconium vs. Titanium grade IV dental implants	Survival rate	98.9% vs. 97.8%
		Crestal bone level changes	0.60 mm vs. 0.61 mm
		Success rate	95.8% vs. 02.6%
Ioannidis et al. [37]	Titanium-zirconium vs. Titanium dental implants	Survival rate	100% vs. 100%
		MBL	0.04 mm vs. 0.01 mm
		FMPS	4% vs. 11%
		BoP	13.8% vs. 20%
		Papilla levels	—
Osman et al. [38]	Zirconia vs. titanium dental implants	Survival rate	90.9% vs. 95.8%
		MBL	0.18 mm vs. 0.42 mm
Osman et al. [39]	Zirconia vs. titanium dental implants	Success rate	Missing
		Prosthodontic maintenance events	45 vs. 34
Al-Nawas [40]	Zirconium-titanium vs. titanium dental implants	MBL	0.34 mm vs. 0.31 mm
		Survival rate	98.9% vs. 97.8%
		Success rate	96.6% vs. 94.4%
Cannizzaro et al. [41]	Non-occlusal loading zirconia dental implants vs. conventional loading zirconia dental implants	Success rate	Missing
		MBL	0.7 mm vs. 0.9 mm

**Table 3 materials-15-01979-t003:** Risk of bias definition.

	Bienz et al. [9]	Patil et al. [33]	Koller et al. [34]	Payer et al. [35]	Müller et al. [36]	Ioannidis et al. [37]	Osman et al. [38]	Osman et al. [39]	Al-Nawas [40]	Cannizzaro et al. [41]
Random Sequence generation	Low	Low	Low	Low	Low	Low	Low	Low	Low	Low
Allocation concealment	Low	Low	High	High	Low	Low	Low	Low	Low	Low
Blinding of participant and personnel	High	High	High	High	Low	High	High	High	Low	High
Blinding of outcome data	High	Low	High	High	Low	High	High	High	Low	High
Selective reporting	High	High	High	Low	High	High	High	Low	High	Low
Other bias	Low	Low	Low	Low	Low	Low	Low	Low	Low	Low

## Data Availability

Data are available on request to CA.

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
