# Peer review of "Endosseous Dental Implant Materials and Clinical Outcomes of Different Alloys: A Systematic Review"

_materials, 2022, doi:10.3390/ma15051979_

Round 1
Reviewer 1 Report
COMMENTS FOR AUTHORS:
- The last ARROW (in Included part) is STILL MISSING in Figure 1. PRISMA flow chart.
- Lines-225&226: "The synthesis was done manually by the authors, any doubts were clarified by an expert author (M.C.) and revised by a native English author (T.F.T.)." This statement can be shifted to the "Author Contributions" column.
- Author can add one reference "....a porous or dense consistency, granular powder and preformed blocks. [Lee, H., Jang, T. S., Song, J., Kim, H. E., & Jung, H. D. (2017). The production of porous hydroxyapatite scaffolds with graded porosity by sequential freeze-casting. Materials, 10(4), 367.]
- Author can add one reference "....by hydroxyapatite crystals which favors migration and the proliferation of some cells, including osteoblasts"[Razali, N. M., Pramanik, S., Osman, N. A., Radzi, Z., & Pingguan-Murphy, B. (2016). Conversion of calcite from cockle shells to bioactive nanorod hydroxyapatite for biomedical applications. J. Ceram. Process. Res, 17, 699-706.].
- All the chemical formula must be written proper; e.g. (HCl/H2â‚‚SO4â‚„), in HCL, 'L' should be small letter 'l'. the H2SO4 should be written as Hâ‚‚SO4â‚„ (H subscript2 S O subscript4), 'TiO2' should be written as 'Ti O subscript2'; similarly Si2O, N2O, P2O2 , Y2O3, CeO2, ZrO2, Al2O3, and etc also should be changed.
Author Response
- The last ARROW (in Included part) is STILL MISSING in Figure 1. PRISMA flow chart.
Dear Reviewer, thank You for Your kind note, the last arrow is present in the PRISMA flow chart.
- Lines-225&226: "The synthesis was done manually by the authors, any doubts were clarified by an expert author (M.C.) and revised by a native English author (T.F.T.)." This statement can be shifted to the "Author Contributions" column.
Dear Reviewer, thank You for Your note, this section has been moved as requested.
- Author can add one reference "....a porous or dense consistency, granular powder and preformed blocks. [Lee, H., Jang, T. S., Song, J., Kim, H. E., & Jung, H. D. (2017). The production of porous hydroxyapatite scaffolds with graded porosity by sequential freeze-casting. Materials, 10(4), 367.]
- Author can add one reference "....by hydroxyapatite crystals which favors migration and the proliferation of some cells, including osteoblasts"[Razali, N. M., Pramanik, S., Osman, N. A., Radzi, Z., & Pingguan-Murphy, B. (2016). Conversion of calcite from cockle shells to bioactive nanorod hydroxyapatite for biomedical applications. J. Ceram. Process. Res, 17, 699-706.].
Dear Reviewer, thank You for Your suggestions, these references have been addes as requested.
- All the chemical formula must be written proper; e.g. (HCl/H2â‚‚SO4â‚„), in HCL, 'L' should be small letter 'l'. the H2SO4 should be written as Hâ‚‚SO4â‚„ (H subscript2 S O subscript4), 'TiO2' should be written as 'Ti O subscript2'; similarly Si2O, N2O, P2O2 , Y2O3, CeO2, ZrO2, Al2O3, and etc also should be changed.
Dear Reviewer, Thank You for Your kind and constructive comment,
Kind Regards
Reviewer 2 Report
The paper can be accepted for publication
Author Response
Thank You for Your kind approval,
Kind Regards
Reviewer 3 Report
This paper aims to describe the state of the art in the field of endosseus dental implants. More precisely, authors compare the use of two different materials (alloys) in the development of implants. The “gold standard” material (based on grade IV titanium) and those made of zirconia.
It is difficult to assess for this referee the quality of the work, as the version that this referee has read appears to be a draft. It seems to be created with the “Word” program with track changes. All along the manuscript some words and paragraphs are cross out. Thus, final version of the manuscript could include different information.
Authors extensively explain the procedure that they have followed in order to select the Clinical trials that they have finally selected to write the paper. At the end, the paper summarizes the results obtained in 10 clinical trials, that full fill all the required fields proposed by the authors. Obviously, it would have been desirable to have a higher number of clinical trials to include in the review, but the ones presented are relevant, and enough to reach the main conclusion (that there are no significant differences between the use of both implants).
The paper is well written, and could result of the interest of researchers and dentist.
I have some suggestions for the consideration of the authors:
- The title of the paper is focused in the description of the review performed using the Randomized Clinical Trials, but in fact, a significant part of the work (most part of the Discussion section) is dedicated to “Implant alloys and surface treatments”, that are not related to the Clinical Trials. Perhaps the title and the structure of the paper could reflect this part (because the Discussion is not related to the Randomized Clinical Trials).
- Even though the paper is based on a PRISMA review, readers would greatly appreciate some images/schemes that could simplified the comprehension of the text. As an example, authors could include a scheme about the process described in page 2 lines 74-78 . “Bone neoformation follows all the stages that characterize direct ossification: arrival of osteoblasts, deposition of osteoid tissue, formation of immature bone with interwoven fibers. The primitive immature bone is progressively resorbed and replaced by mature lamellar bone; this process then leads to the formation of bone around the inserted implant”.
- Another thing authors could include, are some images from the papers they mentioned (obviously for that purpose of course they should ask for permission of the original articles).
- Apparently, as mentioned in Figure 1, some clinical studies have not been considered in the ”present study” due to the lack of access to the paper. This fact can alter the conclusions obtained by the authors, as 287 papers have been discarded arguing “Not accessible text, title, or abstracts or nor full text or not according inclusion criteria”. Authors could ask directly this works to the corresponding authors of the original studies.
- The 3.4 “Results of syntheses” section of the review seems to be a mere description of the results obtained in those 10 papers whose results are already included in table 2. A review is expected to present a mature analysis of an expert in the field highlighting, comparing and analysing the main findings of the original papers.
There are some minor comments:
- Page 2 Line 67. “Around an implant there is always an empty, micrometric space a thin empty layer in which complex biological phenomena take place.
- Please, describe those complex biological phenomena
- Page 2 Line 58. “Osseointegration is defined as the “contact between the vital bone tissue and the surface of a dental implant, without the interposition of soft tissue”.
- The phrase is cut in the draft.
- Page 2 Line 92. Please, consider to introduce a new paragraph when starting the description of zirconia.
- Page 2 Line 97. Is it possible to include the modulus of elasticity for the zirconia, as previously provided for the titanium?.
- Page 2 Line 98. The word “the” al the end of the phrase should be deleted.
- Page 3 Line 110. This referee suggest changing the term “study” for the previous term “review”.
- Page 3 Line 139. This exclusion criteria seems to be redundant. “Studies about dental implant rehabilitation in patients with local or systemic contraindication to dental implant therapy”. If there is a specific contraindication to dental implant therapy, it is difficult to find an implant in the patient.
- Page 8 Table 1. The “methodology” employed in the work of Payer is not included in the table.
- Page 9 Table 2. In some cases, it is not clear to which group of patients corresponds the data present in the fourth column (for example in the work of Bienze et al).
- Page 9 Table 2. In the work of Ioannidis et al. the “Papilla levels” are missing in the fourth column
- Page 13 Table 3. In the table it appears “Blinding of 13articipant and personnel”.
- Page 19 Line 636. Please consider to change the term “aptitude” to tendency or any other synonym in the conclusion (“it is necessary to improve these materials as they have shown a greater aptitude to present fractures”).
- Page 23 Line 808. Please, revise the format of reference 70
Author Response
This paper aims to describe the state of the art in the field of endosseus dental implants. More precisely, authors compare the use of two different materials (alloys) in the development of implants. The “gold standard” material (based on grade IV titanium) and those made of zirconia.
It is difficult to assess for this referee the quality of the work, as the version that this referee has read appears to be a draft. It seems to be created with the “Word” program with track changes. All along the manuscript some words and paragraphs are cross out. Thus, final version of the manuscript could include different information.
Authors extensively explain the procedure that they have followed in order to select the Clinical trials that they have finally selected to write the paper. At the end, the paper summarizes the results obtained in 10 clinical trials, that full fill all the required fields proposed by the authors. Obviously, it would have been desirable to have a higher number of clinical trials to include in the review, but the ones presented are relevant, and enough to reach the main conclusion (that there are no significant differences between the use of both implants).
The paper is well written, and could result of the interest of researchers and dentist.
Dear Reviewer, thank You for Your time and for Your help to improve our manuscript, this is a second stage review as You noted.
I have some suggestions for the consideration of the authors:
- The title of the paper is focused in the description of the review performed using the Randomized Clinical Trials, but in fact, a significant part of the work (most part of the Discussion section) is dedicated to “Implant alloys and surface treatments”, that are not related to the Clinical Trials. Perhaps the title and the structure of the paper could reflect this part (because the Discussion is not related to the Randomized Clinical Trials).
Dear Reviewer, thank You for Your kind suggestion, the manuscript has been modified and the title is suitable now.
- Even though the paper is based on a PRISMA review, readers would greatly appreciate some images/schemes that could simplified the comprehension of the text. As an example, authors could include a scheme about the process described in page 2 lines 74-78 . “Bone neoformation follows all the stages that characterize direct ossification: arrival of osteoblasts, deposition of osteoid tissue, formation of immature bone with interwoven fibers. The primitive immature bone is progressively resorbed and replaced by mature lamellar bone; this process then leads to the formation of bone around the inserted implant”.
Dear Reviewer, thank You for Your suggestion, a figure with this description has been added.
- Another thing authors could include, are some images from the papers they mentioned (obviously for that purpose of course they should ask for permission of the original articles).
Dear Reviewer, we appreciate Your kind suggestion but we think that it is very hard to obtain permission for these digital contents.
- Apparently, as mentioned in Figure 1, some clinical studies have not been considered in the ”present study” due to the lack of access to the paper. This fact can alter the conclusions obtained by the authors, as 287 papers have been discarded arguing “Not accessible text, title, or abstracts or nor full text or not according inclusion criteria”. Authors could ask directly this works to the corresponding authors of the original studies.
Dear Reviewer, thank You for Your kind advice, this is a PRISMA flow chart template sentence, in fact, many of those manuscripts have been deleted as not in accordance with inclusion criteria (paragraph 2.1).
- The 3.4 “Results of syntheses” section of the review seems to be a mere description of the results obtained in those 10 papers whose results are already included in table 2. A review is expected to present a mature analysis of an expert in the field highlighting, comparing and analysing the main findings of the original papers.
Dear Reviewer, thank You for Your kind suggestion, this section has been updated and all RCTs results have been criticized at the end of discussion section.
There are some minor comments:
- Page 2 Line 67. “Around an implant there is always an empty, micrometric space a thin empty layer in which complex biological phenomena take place.
- Please, describe those complex biological phenomena
- Page 2 Line 58. “Osseointegration is defined as the “contact between the vital bone tissue and the surface of a dental implant, without the interposition of soft tissue”.
- The phrase is cut in the draft.
- Page 2 Line 92. Please, consider to introduce a new paragraph when starting the description of zirconia.
- Page 2 Line 97. Is it possible to include the modulus of elasticity for the zirconia, as previously provided for the titanium?.
- Page 2 Line 98. The word “the” al the end of the phrase should be deleted.
- Page 3 Line 110. This referee suggest changing the term “study” for the previous term “review”.
- Page 3 Line 139. This exclusion criteria seems to be redundant. “Studies about dental implant rehabilitation in patients with local or systemic contraindication to dental implant therapy”. If there is a specific contraindication to dental implant therapy, it is difficult to find an implant in the patient.
- Page 8 Table 1. The “methodology” employed in the work of Payer is not included in the table.
- Page 9 Table 2. In some cases, it is not clear to which group of patients corresponds the data present in the fourth column (for example in the work of Bienze et al).
- Page 9 Table 2. In the work of Ioannidis et al. the “Papilla levels” are missing in the fourth column
- Page 13 Table 3. In the table it appears “Blinding of 13articipant and personnel”.
- Page 19 Line 636. Please consider to change the term “aptitude” to tendency or any other synonym in the conclusion (“it is necessary to improve these materials as they have shown a greater aptitude to present fractures”).
- Page 23 Line 808. Please, revise the format of reference 70
Dear Reviewer, thank You for Your specific notes, manuscript has been modified as requested by You.
Kind Regards
Round 2
Reviewer 3 Report
Thank you for taking into consideration the suggestions.